# Prediction of on-field rugby scrummaging contact forces from videos using artificial neural networks

Juan Cordero-Sánchez[1], Zak Sheehy[2], Gil Serrancolí[3,4]\*, Ezio Preatoni[2], Grant Trewartha[5], Dario Cazzola[2]

1 Department of Physiotherapy, Faculty of Medicine and Health Science, University of Alcalá, Alcalá de Henares, Spain, 2 Sport, Health and Exercise Science, Department for Health, University of Bath, Bath, United Kingdom, 3 Simulation and Movement Analysis Lab (SIMMA Lab), Department of Mechanical Engineering, Universitat Politècnica de Catalunya, Barcelona, Catalonia, Spain, 4 Institut de Robòtica i Informàtica Industrial (CSIC-UPC), Barcelona, Catalonia, Spain, 5 School of Health & Life Sciences, Teesside University, Middlesbrough, United Kingdom

\* gil.serrancoli@upc.edu

## Abstract

Rugby is a contact sport with a high risk of injury. Scrums are a key component of the game and represent a collision event associated with serious injuries, especially to the cervical spine. Traditional methods for measuring scrum forces rely on both pressure sensors and scrum machines, but these have limitations for on-field monitoring. Thus, this study investigates the feasibility of using a machine learning approach to predict contact forces in rugby scrummaging during real-game situations, based on only anatomical landmarks of players extracted from video footage. A recurrent neural network (RNN) model was trained to predict contact forces from velocity signals. A dataset of eleven rugby teams (22 forward packs, n = 176 players) during on-field scrum engagements was used. Data augmentation techniques using a generative adversarial network and the Mixup technique were applied to increase the size of the training dataset. The RNN was trained using time-dependent data, with 2D trajectories from video landmarks as inputs and force signals from pressure sensors as outputs. The RNN´s prediction results showed good to excellent agreement between the predicted and measured time-dependent normal contact force signals, with a mean correlation of 0.95 ± 0.05. The mean normalized RMSE was 9.3 ± 4.3% and the mean normalized absolute difference in peak force was 6.7 ± 5.5%. This study demonstrated the feasibility of using RNNs to quantify contact forces in rugby scrummaging using only single-camera video, without players wearing sensors. In combination with a 2D human pose estimation model, the ANN trained in this study could support performance analysis, coaching, technique correction, and injury prediction using in-game data.

**Data availability statement:** Data will be available at https://doi.org/10.5281/zenodo.18048888.

**Funding:** This study was partly supported by the Spanish Research Agency through a grant awarded to J.C. and G.S. (PID2023-150189NA-I00) and Universitat Politècnica de Catalunya. The specific roles of these authors are articulated in the 'author contributions' section. The funders had no role in study design, data collection and analysis, decision to publish, or preparation of the manuscript.

**Competing interests:** The authors have declared that no competing interests exist.

## Introduction

Rugby is a contact sport played worldwide, where players frequently engage in multiple collisions throughout a match [1]. As a result, compared to other team sports, rugby carries a higher risk of injury [2]. The potential for injury increases with high impact forces [3], with more than half of all rugby-associated injuries affecting soft tissues [3–6]. This occurs when a player collides against an opponent as it is the case for tackling and scrum events. Scrummaging involves a dynamic engagement phase and a period of sustained pushing [7,8]. Misdirected or repetitive impact scrum engagements are suggested to be related to the largest incidence rates of cervical spine injuries in rugby union [9,10] as the result of extreme neck flexion, with or without rotation or hyperextension of the C4, C5, and C6 vertebrae [3]. Although severe cervical spine injuries are rare [11], their consequences induce substantial life alterations [12].

Several studies focusing on the measurement of scrum forces have been conducted using a scrum machine [8] or with pressure sensors attached to the players' shoulders [13,14]. However, pressure sensors cannot be used for in-game performance monitoring due to the difficulty of having players wear them during the game. In addition, the development and wider adoption of wearable technology (WT) in sports, such as global positioning systems (GPS) and inertial measurement units (IMU), allow the quantification of both position-specific demands [15,16] and collision frequency [17,18] supporting rugby performance and rehabilitation [19–21] under match-like conditions [22]. However, depending on how easily the WT can be attached and how well it fits the body, its comfort features can significantly impact player's performance. In addition, movements of the WT due to collisions and muscle artifacts can lead to measurement inaccuracies [23]. Therefore, a method to quantify rugby player's positions and velocities as well as the contact forces between them, without interfering with their game, would enhance athletic performance, provide coaching support, aid in technique correction and improve injury prediction [24] from in-game data analysis. This method could involve the use of video-based motion analysis to allow the remote quantification of player kinematics without physical sensors. This enables a fully non-invasive estimation of body positions and velocities during real-game situations. Therefore, implementing video-derived kinematic features as input to artificial neural networks (ANNs) represents a powerful alternative to wearable systems for estimating contact forces in rugby scrummaging.

ANNs have the capacity of dealing with complex, high-dimensional and noisy datasets to identify subtle patterns or hidden structures. In addition, in recent years, there has been a rapid advancement in ANNs, driven by increased computational power, what has promoted their use as a common tool in biomechanics [25]. Consequently, their application in contact force prediction in rugby biomechanics, where the understanding of player movement and contact forces is critical for performance optimization and injury prevention [26], would be valuable.

Therefore, ANNs offer a powerful approach to improve decision-making [27], player performance [28,29] and reduce injury risk [30,31] in rugby. Furthermore, ANNs have been used to predict ground reaction forces (GRF) based on kinematics

during walking [32,33] and running [34–37]. This enables GRF estimation outside of laboratory settings, without the need for experimental techniques typically required to record dynamics data (e.g., force plates). Similarly, predicting shoulder contact forces during rugby scrummaging could become feasible using video-based motion kinematics, just as GRF can be predicted from wearable sensor data during running.

State-of-the-art techniques suggest that collision contact forces in rugby can also be predicted from the kinematics of players. Stitt et al. [38] developed a machine learning model to classify rugby player head impact, while Pomarat et al. [39] trained an ANN to estimate single-player GRF in rugby scrummaging from inertial and force data recorded with instrumented insoles. However, these models were implemented in laboratory settings, so their applicability is limited for real game environments. Thus, this study aims to develop an ANN-based tool to predict shoulder contact forces between individual players from single-camera videos of in-game rugby scrummaging, offering a solution for real-time biomechanical analyses.

## Methods

### Dataset

The dataset used in this study was previously collected and published by Cazzola et al. [13] within the project *Biomechanics of the Rugby Scrum* funded by the International Rugby Board. It contains data of 176 elite rugby players from eleven professional teams (22 forward packs of 8 players: 853.9±28.0 kg per pack). Data collection from these participants was carried out following the institutional ethics committee at the University of Bath, School for Health Research Ethics Approval Panel, and after individual written informed consent was provided by each participant. This dataset consisted of top-view videos and raw kinetic data collected during on-field contested scrum engagements between professional rugby teams. To boost the generalization of the ANN´s model, three different engagement techniques were included in the dataset. Each team (two forward packs, referred to as 'Team A' and 'Team B') performed between 12 and 16 scrums (resulting in a total of 330 scrummaging trials) using the engagement techniques of crouch-touch-pause-engage (CTPE), crouch-touch-set (CTS), and PreBind, in which props prebind with the opposition prior to the 'Set' command. Players' movements were recorded from a top-view camera (HVR-Z5, Sony, Japan) at 50 Hz. Scrummaging contact forces were estimated from pre-calibrated [40] pressure sensors (F-Scan, Tekscan Inc, USA, 500 Hz) attached to the left and right shoulders of the front-row players from 'Team A' (Fig 1).

### Data analysis

The two-dimensional (2D) coordinates of the C7 vertebra and lumbar locations of the front row players were manually labelled from the provided top-view videos. Reliability of these labelled data was assessed and was high, as reported by Preatoni et al. [41]. In addition, in this study, the 2D coordinates and the raw contact forces previously obtained from pressure sensors were low-pass filtered with a second-order Butterworth filter with a cut-off frequency of 20 and 100 Hz, respectively. The components of the midpoint velocity between the C7 and lumbar top-plane locations were calculated as the first derivative of their position vector, then the module vector of the velocity was computed. Velocity-force profiles were computed for each pair in the A1L-B3, A1R-B3, A2L-B3, A2R-B2, A3L-B2, A3R-B1 player engagements, where 'R' and 'L' indicate the contact forces for the right and left shoulders of the 'Team A' players (Fig 1). The velocity curve of each velocity-force profile was calculated as the mean velocity between each pair of player´s velocities. A total of 42 trials were finally selected from the original dataset after data cleaning, which involved removing empty files and scrum trials in which velocity and contact force peak values fell outside the ranges of 2.5–4.5 m·s⁻¹ and 1–4 kN, respectively. Based on the literature [13], values beyond these thresholds were considered physiologically unrealistic or potentially indicative of technical execution errors during the trials, such as sensor misalignment. S1 Fig (Supporting information) displays the engagement technique and player´s shoulder for each scrum trial in the final dataset. The time period analyzed was between the first

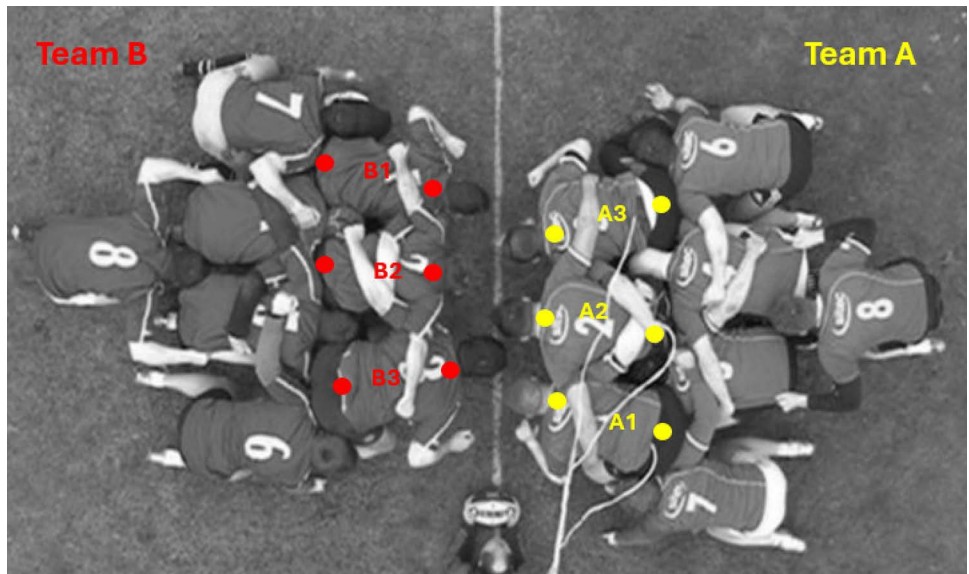

**Fig 1. Camera top view of one scrumming trial showing the vertebrae C7 and lumbar point locations for each 'Team A' and 'Team B' front row player.**

frame when the contact force was higher than 10 N, and the frame with the maximum contact force. Both velocity and contact force signals were time-normalized to 101 data points. Fig 2 shows the distribution of the velocity-force time-dependent curves according to the engagement technique.

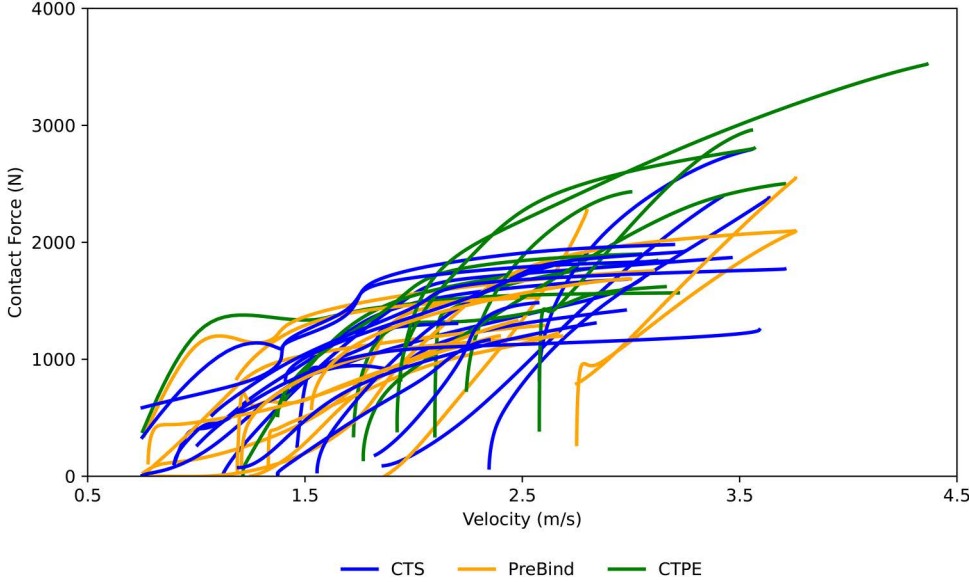

**Fig 2. Velocity-force distribution of all measured time-dependent curves for each engagement technique (CTS, PreBind, CTPE) among all teams and front row players.** Each line represents one player´s shoulder contact trial. This distribution displays the relationship between the linear velocity of the rugby players and the contact force generated during each scrum collision after data cleaning.

## Data augmentation

A generative adversarial network (GAN) was trained to generate one thousand new pairs of velocity-force signals. GANs are generative models that create new samples from scratch. These samples have realistic patterns from the distribution of the original dataset. Therefore, the GAN was trained to synthesize additional realistic velocity–force pairs following the same statistical distribution as the measured data. This data augmentation was implemented to balance the dataset and try to reduce overfitting due to the limited number of real scrum trials, not to modify or substitute the original dataset. The generator was a sequential model with a latent dimension of 50 and composed of one dense layer (26x128 neurons) followed by two convolutional layers (64 and 2 neurons, SELU and tangential activations) ending with one dense layer (202 neurons, tangential activation). Batch normalization layers were implemented in the generator model. The discriminator was a sequential model composed of two convolutional networks (64 and 128 neurons, LeakyReLU activation) and one dense layer (one neuron, sigmoid activation). In addition, layers of dropout were added to regularize the discriminator with a dropout rate of 40%. The convolutional layers had a kernel size of 5 and a stride of 2 with zero padding. The selection of the GAN´s architecture and its hyperparameters relied on manual tuning, according to the correlation and root mean squared error (RMSE) between its generated curves and the measured ones. The resulting final dataset generated by the GAN showed mean correlation and RMSE values of 0.99 and 0.05 m·s$^{-1}$ for the velocity curves, and 0.99 and 128.7 N for the force curves. In both models, the adaptive moment estimation (Adam) algorithm [42] was used as optimizer (with a learning rate of $10^{-3}$) due to its good convergence quality and speed. In addition, using the Mixup technique [43], data were augmented to enlarge the input dataset of the recurrent neural network (RNN). Mixup is a non-generative data augmentation approach that was used to create new velocity–force signal pairs by linearly combining pairs of temporally normalized velocity and force time-series. To construct physically meaningful intermediate time-series while preserving temporal structure a constant mixup coefficient of 0.5 was used. This improves the generalization of the RNN as it provides synthetic data that decreases overfitting.

## Data prediction

A RNN model was trained with the dataset generated by the GAN and enlarged by the Mixup technique. This dataset was split into the training dataset (80%) and the validation dataset (20%). The prediction performance of the RNN was assessed with the test dataset that contained the measured velocity and force data (see the training and test processes in Fig 3). The RNN model was chosen given the sequential and dynamic nature of contact forces, which involves complex temporal dependencies with the velocity signals across time steps. The RNN´s input was the velocity curve of each velocity-force profile, while its output was the predicted force curve of the collision between single player´s shoulders. The RNN was a sequential model composed of two layers with 128 and 64 long short-term memory (LSTM) cells, followed by a dense layer with 1 neuron to produce a single output per time step. The rectified linear unit (ReLU) activation function was used in the last dense layer due to its fast computation and because it does not saturate for positive values. The Adam algorithm, with a learning rate of 0.002 was used as optimizer, while early stopping was implemented to regularize the learning process. The RNN's structure and the learning rate were selected according to the GridSearch algorithm [44] using cross-validation to evaluate all the possible combinations among the number of layers (1, 2 or 3), LSTM cells (32, 64 or 128) and the learning rate (0.001, 0.002 or 0.01). Velocity and force signals were normalized by the maximum absolute value from the entire training dataset to ensure a consistent range of values, thereby facilitating the learning process of the RNN. In the prediction phase, the input data are normalized using the same maximum absolute values determined during training. Pearson´s correlation and RMSE were calculated to evaluate the agreement between the measured and the predicted contact force curves. Correlation was ranked as poor (0–0.5), moderate (0.5–0.75), good (0.75–0.90) and excellent (> 0.9) [45]. Data analysis and neural networks training were carried out in Python (v3.11.5) with TensorFlow and Keras using an Intel i5-14600KF CPU computer with 32GB RAM and a Nvidia GeForce RTX 4060 16GB VRAM GPU. The trained RNN and the data that were used to train and evaluate it, along with a brief tutorial, can be found in this repository: https://doi.org/10.5281/zenodo.18048888.

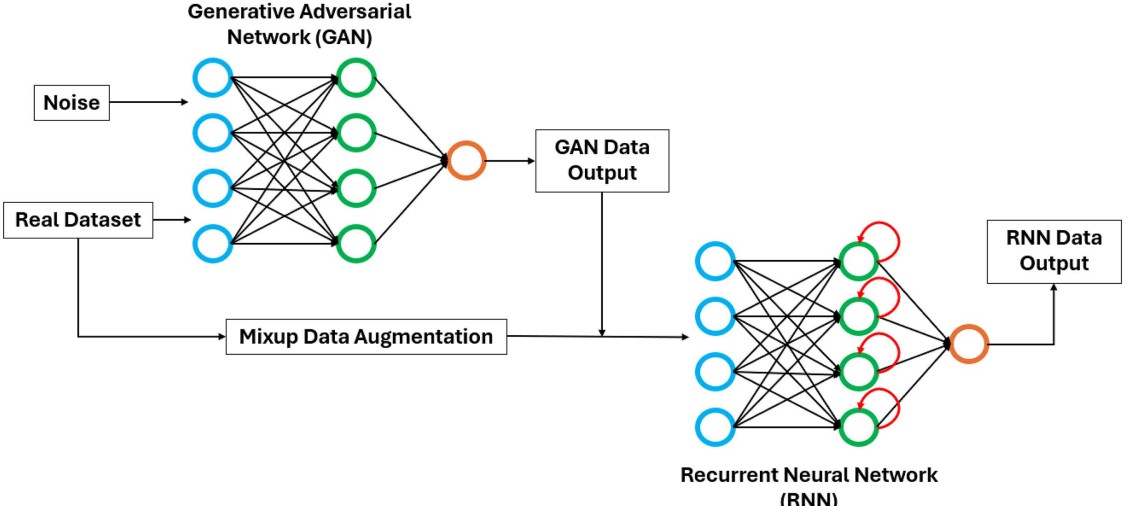

**Fig 3. Diagram of the process to train the RNN to predict contact forces from velocity signals.**

## Results

The agreement between the predicted and measured contact forces ranged from good (0.75–0.90) to excellent (> 0.90) with a mean correlation of 0.95±0.05. The overall mean RMSE and mean absolute peak differences were 327.5±150.8 N (9.3±4.3%) and 236.1±194.2 N (6.7±5.5%), respectively. Comparing the measured and predicted force signals from the start of contact between the front-row players of both teams to the instant the force peak is reached, the highest RMSE values correspond to scrums 12 and 14, which have very similar measured signal curves (Fig 4). In addition, more than half of the force predictions tended to underestimate the measured curves during most of the collision phase. The same occurred with the peak force values, which were more often underestimated by the ANN (see Fig 5 showing the Bland-Altman plot for all peak force values). The mean difference is 54.3 N and the limits are +1.96SD: 640 N and −1.96SD: −530 N. This means that the bias was low, but the large limits indicated high prediction variability. Bland-Altman plot values for each engagement technique were 114.6 N (+1.96SD: 660 N/ −1.96SD: −430 N), −36.9 N (+1.96SD: 510 N/ −1.96SD: −580 N) and 117.6 N (+1.96SD: 730 N/ −1.96SD: −490 N) for the CTPE, CTS and PreBind techniques, respectively.

The major error percentage was found in the PreBind technique for both RMSE (13.5%) and peak difference (10.4%). In the case of player position, the right shoulder of the A3 player reached the highest RMSE (14.5%), while the left shoulder of the A1 player displayed the highest force peak difference (11.4%). Table 1 and Table 2 show the correlation and RMSE values between the measured and predicted contact force curves as well as the peak difference values according to each engagement technique (Table 1) and front row player´s shoulder (Table 2).

## Discussion

In this study, we explored the possibility of estimating engagement forces in a rugby union scrum using only motion (video) data as input to the RNN. Our approach included the use of data augmentation techniques showing promising results in the estimation of engagement forces from velocity data of the front row players in the scrum. These results showed the feasibility of predicting contact forces during scrummaging as well as their respective force impact peaks with mean errors lower than 10% (9.3% and 6.7%, respectively) of the maximum measured contact force.

The proposed work on scrum force estimation enables in-game impact force quantification, unlocking the ability to measure the mechanical load experienced by players during real-time match analyses. This advancement could significantly

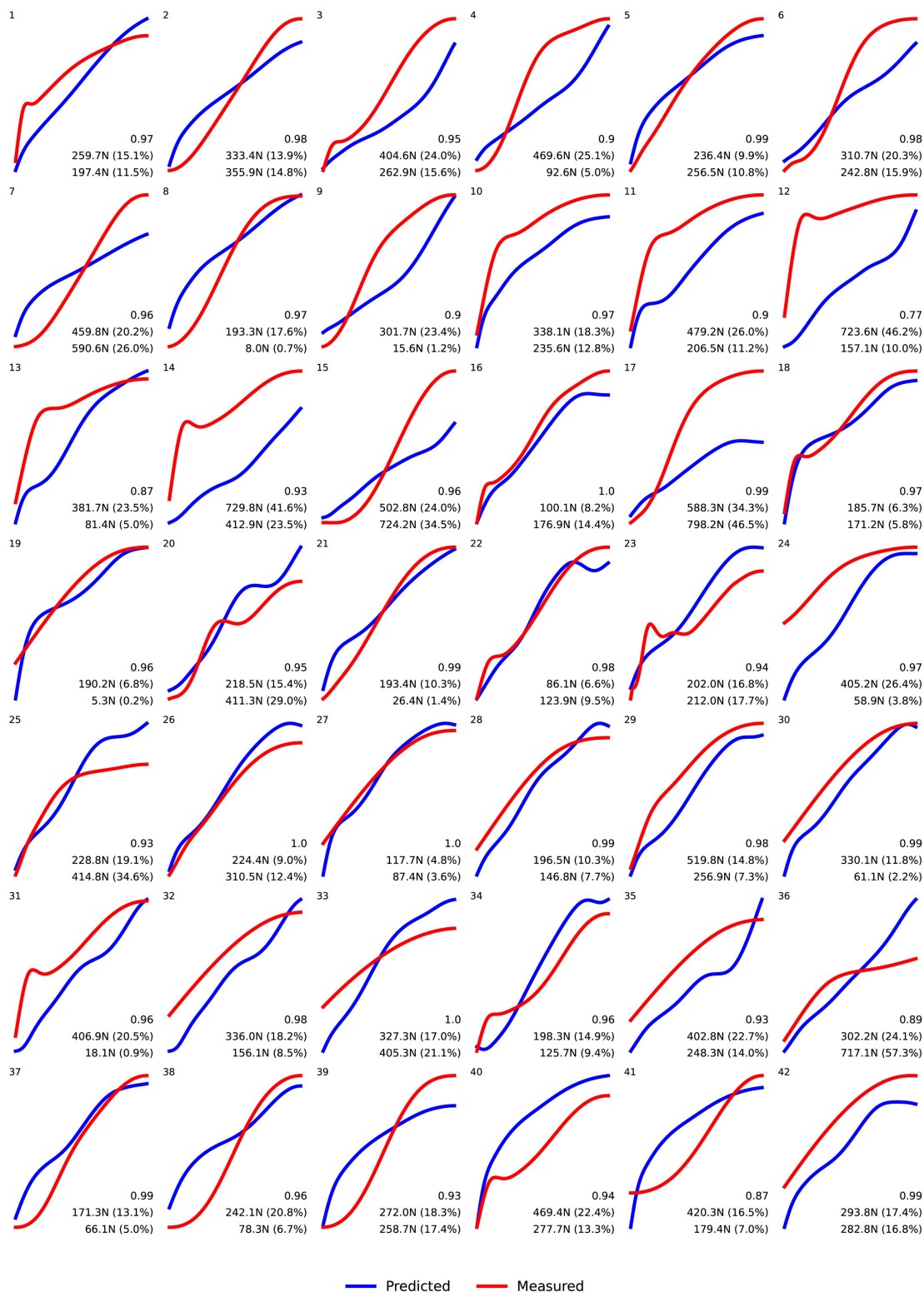

Fig 4. **Predicted (blue) and measured (red) time-normalized contact force curves.** Bottom right values stand for correlation, RMSE (Newtons) and peak difference (Newtons). The error percentage is the RMSE and peak difference normalized to the maximum of the corresponding measured force curve.

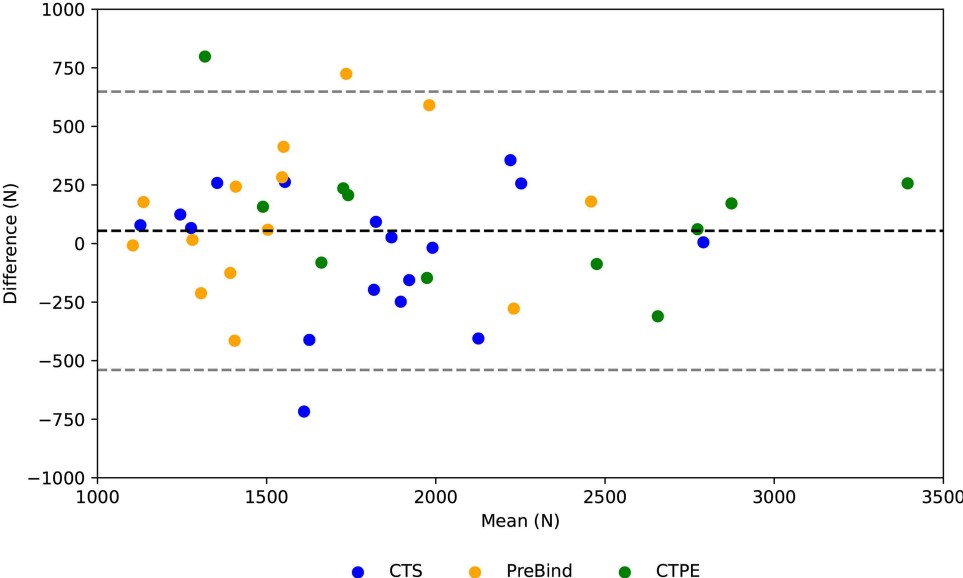

**Fig 5. Bland–Altman plot for all the peak contact force values (Newtons) according to each engagement technique.** The mean difference is 54.3 N and the 95% limits of agreement are +1.96SD: 640 N and −1.96SD: −530 N.

**Table 1. Mean and standard deviation of the correlation, RMSE (Newtons) and absolute force peak difference (Newtons) between measured and predicted contact forces for each engagement technique. The error percentage is the RMSE and peak difference normalized to the maximum of the underlying engagement force data.**

| Engagement | Correlation | RMSE | Peak difference |
|---|---|---|---|
| CTPE | 0.95 ± 0.07 | 371.4 ± 189.4 N (10.5 ± 5.4%) | 228.4 ± 204.3 N (6.5 ± 5.8%) |
| CTS | 0.96 ± 0.03 | 285.5 ± 101.1 N (10.2 ± 3.6%) | 216.5 ± 183.7 N (7.8 ± 6.6%) |
| PreBind | 0.95 ± 0.04 | 344.0 ± 165.4 N (13.5 ± 6.5%) | 265.9 ± 209.4 N (10.4 ± 8.2%) |

**Table 2. Mean and standard deviation of the correlation, RMSE (in Newtons) and absolute force peak difference (in Newtons) between measured and predicted contact forces for each front row player´s shoulder. The error percentage is the RMSE and peak difference normalized to the maximum of the underlying player´s shoulder force data. L and R stand for left and right shoulder, respectively.**

| Player | Correlation | RMSE | Peak difference |
|---|---|---|---|
| A1-L | 0.94 ± 0.05 | 343.5 ± 60.7 N (13.5 ± 2.4%) | 291.4 ± 212.1 N (11.4 ± 8.3%) |
| A1-R | 0.93 ± 0.07 | 458.8 ± 205.0 N (13.0 ± 5.8%) | 260.9 ± 128.6 N (7.4 ± 3.7%) |
| A2-L | 0.96 ± 0.04 | 335.1 ± 141.7 N (11.3 ± 4.8%) | 316.5 ± 248.4 N (10.7 ± 8.4%) |
| A2-R | 0.97 ± 0.03 | 292.8 ± 175.6 N (11.7 ± 7.0%) | 254.5 ± 261.4 N (10.2 ± 10.6%) |
| A3-L | 0.96 ± 0.02 | 241.5 ± 90.0 N (8.6 ± 3.2%) | 176.2 ± 131.7 N (6.3 ± 4.7%) |
| A3-R | 0.95 ± 0.05 | 303.2 ± 130.7 N (14.5 ± 6.3%) | 102.7 ± 97.3 N (4.9 ± 4.7%) |

improve the calculation of rugby performance indicators [29,46,47], facilitate tactical decisions [27,28,48], and ultimately reduce injuries [24,49,50]. In addition, our findings complement other machine learning works focused on both detecting impact events [20,31,51–53] and estimating poses in rugby [30,54]. These machine learning models employed in prior research can provide the time of impact and the player's posture, which are the inputs required for applying our scrum engagement force prediction model. However, as the used dataset to train this model consisted exclusively of elite male rugby players, its generalizability to other populations (e.g., female, youth, or amateur players) remains untested. These groups differ in anthropometry, strength, and scrummaging technique. Nevertheless, as the inputs of our model are solely linear velocities, its predictions should not be strongly affected by players' physical differences. Even so, future work should include additional data from diverse cohorts to enable transfer learning and ensure model applicability across playing levels.

The RNN model developed in this study was trained to estimate the contact force signal during the engagement phase of the force production in scrummaging. This impulsive engagement phase is highly influenced by the mass and the engagement speed of the players [55,56]. Although our RNN did not consider the mass of the players, this would not be a weakness of the RNN since Quarrie and Wilson [56] found that, even though the mean correlation between players pack mass and the scrum force was higher than 0.8, the 95% confidence interval of this correlation included zero making this relationship variable. In addition, it should be noted that heavier forward packs generate greater impact force only in the case of the full scrum technique [57]. Moreover, the prediction accuracy of the RNN developed in this study tended to be similar among the different engagement techniques analysed. Only the PreBind percentage errors of the RNN predictions were slightly higher. Scrummaging collisions result in lower impact forces with the PreBind technique, as demonstrated experimentally by [13]. This could indicate that the lower the scrum impact force, the worse the RNN's accuracy. The mean maximum RMSE and peak absolute differences among players were 217.4 N and 213.8 N, respectively; and among engagement techniques, the corresponding values were 59.0 N and 49.4 N. This shows that the error observed using different engagement techniques and player positions in the front row was lower than the intrinsic error (~250/300N) of the pressure sensors [40].

The novelty of our work is the prediction of contact forces during scrummaging collisions using a RNN combined with video motion capture. Previous research in literature predicting dynamic data from kinematics had primarily focussed on GRF in tasks like gait and running [33,35,58], and to a lesser extent, pedal forces in cycling [59]. Focusing on rugby, Stitt et al. [38] recently developed a head impact classification machine learning model to study the rugby head impact in a laboratory setting. In addition, Pomarat et al. [39] trained a LSTM neural network to predict three-dimensional GRF from pressure insole signals during scrums in laboratory isolated conditions using a rugby scrum machine. Although the sports task was scrum collision, the application and purpose of these studies were different. Therefore, a straightforward comparison with our results was not possible. Additionally, the data used to train our RNN model were collected during match-realistic scrum situations while the output data were the shoulder contact forces. However, these studies could be used in combination with the method proposed here to get a deeper insight into both the risk of head impacts and the dynamics of scrummaging, by performing inverse dynamics analyses knowing the external forces (GRF and shoulder contact forces) and their points of application along with the kinematics of the front-row players.

The force prediction results of our study were consistent with the error magnitudes reported in other studies for different sports tasks ranging between 12.0%−15.0% [33] and 4.0%−14.0% [35] for GRF, and between 9.9%−13.0% [59] for pedal forces. Nevertheless, caution should be taken when comparing our results with other studies due to differences in both the movement tasks and the nature of the external forces being predicted. Most studies predicted GRF during cyclic movements and, as it was highlighted by [60], model performance is strongly dependent on the task that it was trained on. In addition, although scrum shoulder contact forces and GRF are both external forces, they arise from interactions between different types of bodies. Contact forces, generated by collisions between two (or more) bodies, depend on the mechanical properties of those bodies [61]. Shoulder contact forces come from the collision between two soft tissues in movement,

which could add more variability into the force signal. This could be a reason for the results observed in the Bland–Altman plot, whose limits of agreement (±600 N) indicate a notable variability. However, this represents approximately 15% of the maximum measured peak force. Therefore, the model could be better suited for group-level monitoring and trend analysis (e.g., detecting systematic increases in impact load or comparing engagement techniques) rather than for making clinical or safety judgments on single events and individual predictions. Thus, despite scrummaging being a challenging task involving multiple players performing high-impact movements in real-world situations, the RNN trained in this study achieved prediction accuracies, in terms of mean error percentages, comparable to those reported in other studies that predicted external forces during more regular movements under highly controlled conditions.

The main limitation of this study was that the training of the RNN was conducted solely based on 2D body locations obtained from a top-view perspective. This approach resulted in a loss of information regarding the player's poses during the scrum, which could be highly relevant to evaluate the correlation between player kinematics and scrum contact force generation. However, in rugby scrummaging, players are trained and conditioned to maximize forward (horizontal) force production while maintaining stable body positions, with vertical force and movement components playing a secondary role in the overall movement strategy [62]. Nevertheless, there is considerable room for improvement in terms of both the accuracy and robustness of force predictions. Thus, to improve the performance of the RNN developed in this study, future work should focus on estimating the full-body pose of players during the scrum to combine it with scrum collision physics models and ANNs. By integrating 2D (or even 3D) positions of multiple body joints using pose estimation neural network models, it may be possible to extract a much richer set of anatomical landmarks from video data. This extended set of anatomical features would enable the computation of additional kinematic descriptors such as 3D joint angles, velocities and accelerations, which are critical for understanding force transmission through the kinetic chain during scrum engagement.

However, it must be considered that the transitioning from manually labelled landmarks to automated pose estimation introduces additional sources of error related to landmark misidentification, depth ambiguity, and occlusion. Menychtas et al. [63] have shown that 2D pose estimation algorithms yield accurate estimates for large joint movements but struggle to capture small, high-frequency motions or those occurring out of plane. Similarly, automated vision-based tracking systems may exhibit greater variability when the visibility of landmarks is compromised [64]. On the other hand, manual labelling is more consistent, but it could introduce a static offset due to a user´s bias [63]. To mitigate these limitations, future work should employ confidence-weighted filtering to disregard low-certainty keypoints, apply temporal smoothing to correct transient errors, and train pose-estimation models using hybrid datasets combining automatically estimated landmarks with manual corrections. These strategies can improve robustness and help bridge the accuracy gap between manual and automated tracking systems. Nevertheless, there is still a significant time-saving benefit of the automated algorithms over manual annotation [63]. This efficiency facilitates the annotation of a larger number of body landmarks and, consequently, incorporating this richer information as input to the neural network would provide a more comprehensive representation of player´s postures and motions, ultimately enabling more accurate predictions of contact forces. In addition, embedding physical laws such as Newtonian mechanics or even simplified musculoskeletal models directly into the RNNs training process could improve the physical consistency of the contact force predictions. This could ensure that predicted contact forces do not violate biomechanical limits and that they are consistent with observed accelerations.

To conclude, this study demonstrates the feasibility of using RNN models to predict engagement forces in rugby union scrums. This opens the door to real-time, in-game analysis of scrum collisions, which could be applied directly on the field via embedded sensors or computer vision. Such systems could provide instant feedback on force magnitudes, highlighting dangerously high engagements, and detect performance inconsistencies. As a result, coaches and medical staff could use this information during the game to assess injury risk, identify players showing signs of fatigue or poor form, or make tactical adjustments in real time. This level of live analysis would revolutionize player monitoring, adding a data-driven layer of decision-making unavailable during match play, what represents a meaningful step forward in biomechanical analysis of rugby performance.

## Supporting information

**S1 Fig. Matrix of measured scrum trials showing what engagement technique (CTS: blue, PreBind: orange, CTPE: green) and player´s s*houlder each trial comes from. I*t** shows both the engagement technique (CTS: blue, PreBind: orange, CTPE: green) and the players' shoulders (A1, A2, A3; L: left, R: right. See Fig 1) for each measured trial that makes up the RNN test dataset. Trial numbers (upper left) correspond to the same trials shown in Fig 4.
(EPS)

## Author contributions

**Conceptualization:** Juan Cordero-Sánchez, Dario Cazzola.

**Data curation:** Juan Cordero-Sánchez, Zak Sheehy.

**Formal analysis:** Juan Cordero-Sánchez, Zak Sheehy.

**Investigation:** Juan Cordero-Sánchez.

**Methodology:** Juan Cordero-Sánchez, Zak Sheehy.

**Project administration:** Dario Cazzola.

**Resources:** Juan Cordero-Sánchez, Dario Cazzola.

**Supervision:** Gil Serrancolí, Ezio Preatoni, Grant Trewartha, Dario Cazzola.

**Validation:** Juan Cordero-Sánchez, Gil Serrancolí.

**Visualization:** Juan Cordero-Sánchez.

**Writing – original draft:** Juan Cordero-Sánchez.

**Writing – review & editing:** Juan Cordero-Sánchez, Gil Serrancolí, Ezio Preatoni, Grant Trewartha, Dario Cazzola.

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
