## [Decision Letter · Decision Letter 0]

9 Nov 2025

PONE-D-25-40825Prediction of on-field rugby scrummaging contact forces from videos using Artificial Neural Networks.PLOS ONE

Dear Dr. Serrancolí,

Thank you for submitting your manuscript to PLOS ONE. After careful consideration, we feel that it has merit but does not fully meet PLOS ONE’s publication criteria as it currently stands. Therefore, we invite you to submit a revised version of the manuscript that addresses the points raised during the review process. This status of major revision is based on the following requirements being met as a minimum: A. Ethical and Editorial Non-Compliance (Fatal Flaws)

1. Severe Violation of the Open Data Policy (PLOS ONE): The authors explicitly state: “The data underlying the results presented in the study are not available. Data are partially owned by a third party and authors do not have permission to share the data.” This violates the fundamental requirement of PLOS ONE that all underlying data be made available without restriction upon publication. The justification of "third-party ownership" is unacceptable for a study seeking to contribute to open science. The reproducibility of the work is, therefore, impossible.

2. Contradiction and Failure in the Ethics Statement: The manuscript marks "N/A" in the Ethics Statement section, implicitly claiming not to involve human participants. However, the Methods section confirms the use of data from 176 elite players with written informed consent and approval from the University of Bath Ethics Committee. The omission or deliberately incorrect statement of ethical approval is a serious breach of editorial integrity.

If the above requirements are not met in the revised and resubmitted work, then this paper will be rejected without further review. Please also address the other methodological and reporting comments provided by all reviewers.

We look forward to receiving your revised manuscript.

Kind regards,

Christopher Kirk, PhD

Academic Editor

PLOS ONE

Journal Requirements:

2. Please note that PLOS One has specific guidelines on code sharing for submissions in which author-generated code underpins the findings in the manuscript. In these cases, all author-generated code must be made available without restrictions upon publication of the work. Please review our guidelines at https://journals.plos.org/plosone/s/materials-and-software-sharing#loc-sharing-code and ensure that your code is shared in a way that follows best practice and facilitates reproducibility and reuse.

Reviewers' comments:

Reviewer's Responses to Questions

**Comments to the Author**

1. Is the manuscript technically sound, and do the data support the conclusions?

Reviewer #1: Partly

Reviewer #2: Yes

Reviewer #3: Yes

2. Has the statistical analysis been performed appropriately and rigorously? 

Reviewer #1: Yes

Reviewer #2: Yes

Reviewer #3: Yes

3. Have the authors made all data underlying the findings in their manuscript fully available?

Reviewer #1: No

Reviewer #2: No

Reviewer #3: No

4. Is the manuscript presented in an intelligible fashion and written in standard English?

Reviewer #1: Yes

Reviewer #2: Yes

Reviewer #3: Yes

5. Review Comments to the Author

Reviewer #1: This research implementing machine learning technique to predict the injury risk in the high-risk sports, rugby, to provide further information for team staff to avoid injury risk. I have several comments as below:

1. From line 55-71, authors describe the traditional data collection shall be improve by implementing wearable technology (WT) include GPS and IMU. However, it seems that this article tried to improve the prediction performance by integrating video-analysis techniques. I suggest to revise this paragraph to enhance the importance of implementing video-based analysis instead of highlights the WT importance.

2. From line 80, authors raise the predict target GRF. However, it shall be clarified that GRF is stand for “Ground Contact Forces” or “Ground Reaction Forces.”

3. On line 56-59, authors illustrate that the traditional methods “miss forces transferred through neck and head…..” However, from my point of view, this research also used the sensor installed on player’s shoulder. How does this work is different from other research?

4. On line 97-111, dataset shall be describe as “how many total dataset had been detected.” That is, how many scrummaging had been performance by the 176 elite rugby players to produce the numbers of data collected.

5. Continued with No.4, if the dataset had been modified, what is the reason for implementing GAN to increase the data size?

6. The variables used in this research shall be identified. This research aims to predict the injury risk. However, using of variables represents “injury risk” is not clear enough.

Reviewer #2: Strengths

The study effectively addresses a genuine gap by employing machine learning to quantify rugby scrummaging forces in real-world settings. The dataset is both large and of high quality, encompassing 11 elite teams, thereby lending substantial weight to the analysis and results. The application of GAN and Mixup augmentation, coupled with meticulous hyperparameter tuning, reflects a thoughtful and contemporary approach to model development. The reported results are robust, exhibiting high correlation and relatively low error margins. The breakdown by technique and position is particularly valuable and enhances the study's credibility. The practical implications for non-invasive player monitoring and injury prevention are well-articulated and potentially impactful.

Weaknesses and Suggestions for Improvement

Methods detail and replicability

The description of manual labeling of C7 and lumbar landmarks is insufficient. It is essential to specify the number of raters involved, whether inter- or intra-rater reliability was assessed, and the software utilized, as manual labeling is susceptible to error. Data cleaning is mentioned, but the number of trials excluded and the justification for the chosen thresholds are not reported. Without this information, it is challenging to assess whether exclusions biased the dataset. The GAN and Mixup approaches are introduced but not described in sufficient detail for replication. A clearer specification of the GAN architecture and training procedure, as well as the parameters used in Mixup, is necessary.

Limitations of 2D data

Although the limitation of relying on a single top-view camera is acknowledged, this point warrants more explicit discussion. Important postural information is inevitably lost in 2D, which could directly account for some of the observed prediction variability.

Prediction variability

The Bland–Altman plots demonstrate wide limits of agreement (approximately ±600 N). This is not trivial, particularly when considering injury risk thresholds. While the average error is low, a clearer discussion of the practical implications of such variability would benefit readers—for instance, whether the tool is more suited for monitoring trends across groups rather than making judgments on individual events.

Automated pose estimation

The manuscript suggests eventual automation via pose estimation, yet the challenges of this transition are not discussed. It is important to comment on the likely decrease in accuracy when transitioning from carefully labeled manual landmarks to automatically detected ones, and how this issue might be addressed.

Generalizability

The dataset utilized in this study is derived from elite male rugby teams. It is pertinent to briefly consider the extent to which the model may be applicable to other cohorts, such as youth, amateur, or female players, where variations in anthropometry and technique are evident.

Presentation Issues

The tables and figures are generally well-presented; however, it is advisable to include units and sample sizes in the captions where relevant. Additionally, Table 2 appears to have a formatting issue in the A3-R row, which requires correction. It may also be beneficial to incorporate a small set of example traces (best, average, worst prediction) to demonstrate the model's performance under varying conditions.

Recommendation

Overall, this manuscript is robust and offers a significant contribution to the field of applied sports biomechanics. The novelty, quality of the dataset, and clarity of presentation are commendable.

Nonetheless, prior to publication, I recommend minor revisions to enhance the reproducibility of the methods and to provide a more nuanced discussion of the study's limitations and implications.

Reviewer #3: Critical Review Letter – Manuscript PONE-D-25-40825

Title: “Prediction of on-field rugby scrummaging contact forces from videos using Artificial Neural Networks”.

Dear Researchers,

Thank you for the invitation to review this manuscript. The study presents an innovative proposal of high clinical and sporting relevance: predicting contact forces in the rugby scrum using only video kinematics and Recurrent Neural Networks (RNNs). The potential for cervical injury prevention is promising. However, after a detailed analysis of the submission, I have identified critical issues and severe ethical and editorial failures that fundamentally compromise the validity, reproducibility, and compliance of the work.

Final Recommendation: Rejection with the possibility of resubmission only after substantial revision and proof of compliance with editorial policies.

The severity of the flaws is categorized below:

A. Ethical and Editorial Non-Compliance (Fatal Flaws)

1. Severe Violation of the Open Data Policy (PLOS ONE): The authors explicitly state: “The data underlying the results presented in the study are not available. Data are partially owned by a third party and authors do not have permission to share the data.” This violates the fundamental requirement of PLOS ONE that all underlying data be made available without restriction upon publication. The justification of "third-party ownership" is unacceptable for a study seeking to contribute to open science. The reproducibility of the work is, therefore, impossible.

2. Contradiction and Failure in the Ethics Statement: The manuscript marks "N/A" in the Ethics Statement section, implicitly claiming not to involve human participants. However, the Methods section confirms the use of data from 176 elite players with written informed consent and approval from the University of Bath Ethics Committee. The omission or deliberately incorrect statement of ethical approval is a serious breach of editorial integrity.

B. Methodological and Technical Flaws (ML and Biomechanics)

3. Questionable Data Augmentation Methodology (GAN/Mixup): The use of a Generative Adversarial Network (GAN) to create 1,000 synthetic force signals in a safety context has not been adequately justified in terms of physical validity. The high statistical correlation between generated and measured data (R=0.99) does not guarantee that the synthetic forces respect the biomechanical laws of a human collision. The risk of the trained model learning synthetic artifacts from the GAN, instead of real patterns, is high and dangerous for an application aimed at preventing injuries.

4. Non-Scalable Methodology: The model relies on manual 2D labeling of the C7 and Lumbar landmarks. This makes real-time, on-field (in-game) application unfeasible and burdensome. The authors do not provide: a) A labeling protocol, b) Inter-rater reliability metrics (essential for validating the kinematic input), or c) A validated automated pose estimation model to replace manual labeling.

5. Clinically Unacceptable Variability in Results: Despite the low normalized RMSE (9.3%), the Bland-Altman analysis reveals extreme Limits of Agreement (–530 N to +640 N). In a high-energy collision scenario, this margin of error in peak force prediction is clinically unacceptable and can mask the difference between a safe scrum and one with a high risk of cervical injury.

6. Omission of Critical Biomechanical Covariates: The model ignores the individual body mass of the players as an input variable. Contact forces in a collision are a direct function of mass and acceleration (F=ma). The exclusion of this fundamental biomechanical variable compromises the model's robustness and is a likely cause of the high variability observed in the Bland-Altman analysis (Point 5).

C. Presentation and References

7. Exaggerated Conclusion: The claim that the method "could revolutionize player monitoring" is highly speculative. The unacceptable clinical variability (± 640 N), the failure of scalability (manual labeling), and the dependence on non-sharable data prevent the work from being considered a transformative tool in its current state.

8. References (Requires Review): Excessive self-citation (20% of the total) and the use of conference proceedings (e.g., Pomarat et al., [39]) to substantiate methodological arguments were identified, which weakens the theoretical basis of the submission.

Summary: The study has significant intellectual potential but is irremediably compromised by ethical failures (Data and Consent) and methodological flaws (Scalability and Clinical Validity of Error). Rejection is the only possible recommendation.

6. PLOS authors have the option to publish the peer review history of their article (what does this mean? ). If published, this will include your full peer review and any attached files.

**Do you want your identity to be public for this peer review?** For information about this choice, including consent withdrawal, please see our Privacy Policy .

Reviewer #1: No

Reviewer #2: **Yes:** Dr. SUKUMARAN, C

Reviewer #3: **Yes:** João Carlos Alves Bueno

---

## [Author Response · Author response to Decision Letter 1]

13 Jan 2026

We thank the reviewers for providing feedback on our paper. We trust that changes resulting from your comments have improved the quality of this manuscript. All comments have been addressed within the file Response to Reviewers. Changes to the manuscript have been highlighted in red in the Manuscript with Track Changes.

---

## [Decision Letter · Decision Letter 1]

24 Feb 2026

Prediction of on-field rugby scrummaging contact forces from videos using Artificial Neural Networks.

PONE-D-25-40825R1

Dear Dr. Serrancolí,

We’re pleased to inform you that your manuscript has been judged scientifically suitable for publication and will be formally accepted for publication once it meets all outstanding technical requirements.

Kind regards and congratulations,

Christopher Kirk, PhD

Academic Editor

PLOS One

Additional Editor Comments (optional):

Reviewers' comments:

Reviewer's Responses to Questions

**Comments to the Author**

1. If the authors have adequately addressed your comments raised in a previous round of review and you feel that this manuscript is now acceptable for publication, you may indicate that here to bypass the “Comments to the Author” section, enter your conflict of interest statement in the “Confidential to Editor” section, and submit your "Accept" recommendation.

Reviewer #1: All comments have been addressed

Reviewer #2: All comments have been addressed

2. Is the manuscript technically sound, and do the data support the conclusions?

Reviewer #1: Yes

Reviewer #2: Yes

3. Has the statistical analysis been performed appropriately and rigorously? 

Reviewer #1: Yes

Reviewer #2: Yes

4. Have the authors made all data underlying the findings in their manuscript fully available?

Reviewer #1: Yes

Reviewer #2: Yes

5. Is the manuscript presented in an intelligible fashion and written in standard English?

Reviewer #1: Yes

Reviewer #2: Yes

6. Review Comments to the Author

Reviewer #1: The authors had appropriately response my previous comments. I have no further comments and leave the decision back to editorial office.

Reviewer #2: The authors have adequately addressed all comments raised in the previous review. The revised manuscript provides sufficient methodological detail to ensure reproducibility, including clear descriptions of manual labeling, data cleaning, and data augmentation procedures. The discussion of limitations related to 2D data, prediction variability, automated pose estimation, and generalizability has been appropriately expanded and balanced.

The previously noted formatting issue in Table 2 has been corrected, and the overall presentation is clear. No substantive concerns remain, and the manuscript is suitable for publication.

7. PLOS authors have the option to publish the peer review history of their article (what does this mean? ). If published, this will include your full peer review and any attached files.

**Do you want your identity to be public for this peer review?** For information about this choice, including consent withdrawal, please see our Privacy Policy .

Reviewer #1: No

Reviewer #2: No
